# RECTIFIED SPARSE ATTENTION FOR EFFICIENT LONG-SEQUENCE GENERATION

## ABSTRACT

Efficient long-sequence generation is a critical challenge for Large Language Models. While recent sparse decoding methods improve efficiency, they suffer from KV cache misalignment, where approximation errors accumulate and degrade generation quality. In this work, we propose Rectified Sparse Attention (ReSA), a simple yet effective method that combines block-sparse attention with periodic dense rectification. By refreshing the KV cache at fixed intervals using a dense forward pass, ReSA bounds error accumulation and preserves alignment with the pretraining distribution. Experiments across math reasoning, language modeling, and retrieval tasks demonstrate that ReSA achieves near-lossless generation quality with significantly improved efficiency. Notably, ReSA delivers up to 3.77× end-to-end speedup under decoding at 256K sequence length, making it a practical solution for scalable long-context inference.

## 1 INTRODUCTION

The ability to process long contexts has become a core requirement for Large Language Models, with context lengths up to millions of tokens (Reid et al., 2024; Yang et al., 2025). In particular, long sequence generation has received growing attention, especially due to the demand for test-time scaling (Guo et al., 2025; Jaech et al., 2024).

Despite this progress, efficient long-sequence generation remains a significant challenge. In standard autoregressive decoding, each token must attend to the full KV cache, leading to frequent memory access and increased IO pressure. This bottleneck severely limits throughput, especially in long-context scenarios where memory access dominates latency.

Recent works (Liu et al., 2024; Tang et al., 2024) used sparse decoding to alleviate this issue, which selectively attends to a subset of the context, achieving accuracy comparable to dense attention on long inputs while reducing computational cost. However, as shown in Figure 1, they often suffer from worse performance with increasing length. Since **computation errors accumulate in the KV cache during sparse decoding**, the attention computation suffers from the misalignment between training and inference, contributing to performance degradation.

In this work, we propose Rectified Sparse Attention (ReSA), a simple yet effective approach that achieves near-lossless long-sequence generation quality while maintaining high inference efficiency. ReSA leverages block-sparse attention (Tang et al., 2024) for fast retrieval and further improves memory efficiency by applying shared grouping (Yuan et al., 2025), allowing query heads to reuse attention patterns. To address the error accumulation issue, we introduce dense rectification, where the sparse KV cache is periodically refreshed with a parallel dense forward pass. This ensures that approximation errors are bounded within a constant range, preventing long-term degradation.

We conduct experiments to demonstrate the effectiveness of ReSA. On math reasoning benchmarks, ReSA

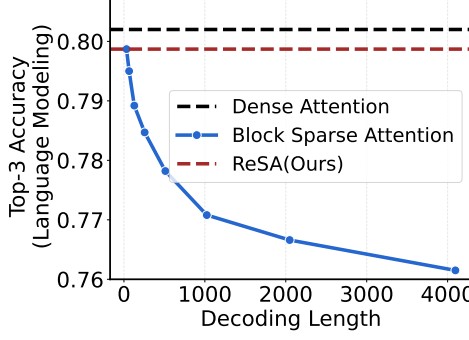

Figure 1: Sparse decoding performance becomes worse with increasing decoding length due to error accumulation of KV cache.

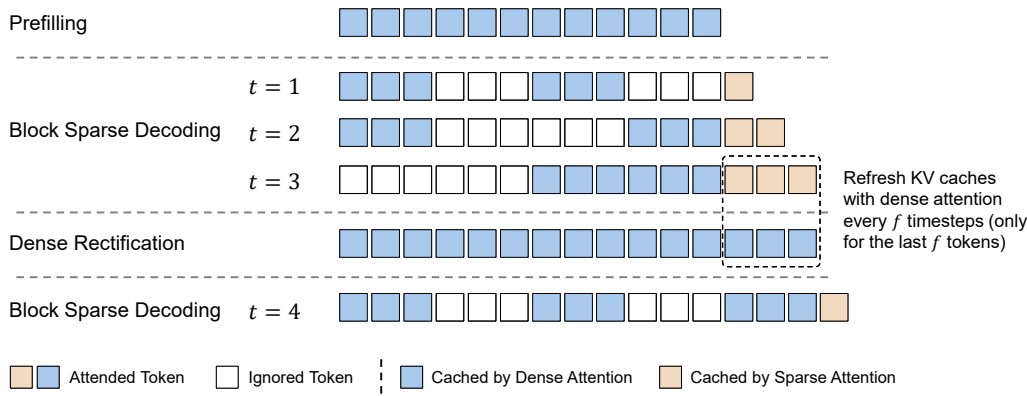

Figure 2: Overview of ReSA. After completing the prefill stage, the model enters sparse decoding. Once the number of generated tokens reaches the rectification frequency, a rectification step is performed to construct a lossless compact KV cache, after which sparse decoding resumes.

achieves strong test-time scaling and matches dense attention in long-sequence settings. In language modeling, ReSA significantly closes the quality gap between sparse and dense decoding. On the efficiency side, our approach yields up to $3.77\times$ end-to-end speedup at 256K context length, showing strong practical utility for real-world deployment.

## 2 RECTIFIED SPARSE ATTENTION

ReSA primarily involves two alternating phases, sparse decoding and periodic rectification. During the decoding phase, we employ the group block sparse attention mechanism, which significantly reduces computational and memory overhead, enabling fast autoregressive inference. During the rectification stage, the decoding tokens are forwarded in parallel to correct approximation errors in KV cache introduced by sparse decoding. By alternating between sparse generation and dense rectification, ReSA enables scalable long-context inference while ensuring the generation quality.

### 2.1 GROUP BLOCK SPARSE ATTENTION

Self-attention mechanisms are the core component of Transformer architectures, enabling each token to attend to all previous tokens.

We adopt a block-sparse attention design that selectively attends to a small number of relevant memory blocks rather than the entire context. Formally, in Group-Query Attention (GQA) (Ainslie et al., 2023), given a sequence of $n$ tokens, the query $Q \in \mathbb{R}^{h \times g \times n \times d}$, key $K \in \mathbb{R}^{h \times n \times d}$, and value $V \in \mathbb{R}^{h \times n \times d}$, the block size $b$ and block sparse mask $M \in \{0, 1\}^{h \times n \times n/b}$, the block-sparse attention is computed as:

$$\text{GBSA}(Q, K, V, M)_{ij} = \text{softmax}\left(\frac{Q_{ij}K_i^\top \cdot \overline{M}_i}{\sqrt{d}}\right)V_i, \ \overline{M}_{ijk} = M_{ij\lfloor k/b \rfloor} \tag{1}$$

GBSA adopts a query-dependent sparsity pattern, where each query attends to a limited set of key blocks determined by $M$. Since each selected key block corresponds to a contiguous memory region in the KV cache, this design ensures both high performance and memory efficiency during inference. Note that we further accelerate decoding by maintaining a shared sparse pattern within each GQA group (Yuan et al., 2025).

**Block Representation**  Following Quest (Tang et al., 2024), we represent the key-value memory using blocks to enable efficient retrieval. Specifically, given a key matrix $k \in \mathbb{R}^{n \times d}$, we partition it into non-overlapping blocks of size $b$, where each block contains $b$ consecutive tokens. For the $i$-th block, we compute two block descriptors:

$$\begin{aligned} k_{\text{block\_min},i} &= \min(k_{ib:(i+1)b}) \\ k_{\text{block\_max},i} &= \max(k_{ib:(i+1)b}) \end{aligned} \tag{2}$$

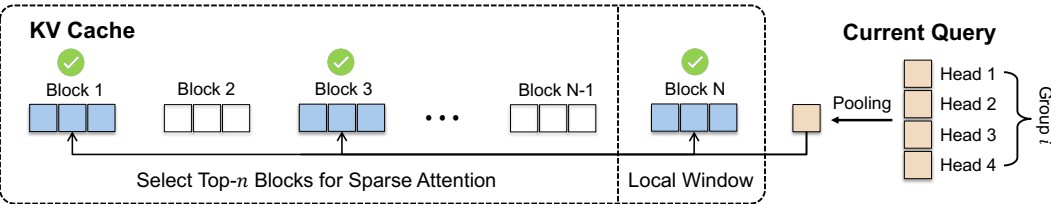

Figure 3: Overview of Group Block Sparse Attention. For each group of query heads, we perform average pooling and enforce the selection of the same KV blocks across all heads within the group.

where $\min(\cdot)$ and $\max(\cdot)$ are applied element-wise across the block dimension.

Notably, the block representation is entirely training-free, relying solely on statistical descriptors. Our method remains compatible with more advanced block representation strategies, such as SeerAttention (Gao et al., 2024), where block keys are fine-tuned jointly with the model to achieve higher retrieval precision if needed.

**Block Selection** During decoding, given a pooling query $q \in \mathbb{R}^d$ for each GQA group and a set of block descriptors $\{(k_{\text{block\_min},i}, k_{\text{block\_max},i})\}_{i=1}^{M}$, we compute similarity scores following the Quest algorithm (Tang et al., 2024). Specifically, the score between the pooling query and block $i$ is calculated as:

$$\text{score}_i = \sum_{j=1}^{d} \max\left(q_j \times (k_{\text{block\_max},i})_j, q_j \times (k_{\text{block\_min},i})_j\right) \tag{3}$$

where $q_j$ denotes the $j$-th dimension of the pooling query, and $(k_{\text{block min},i})_j$, $(k_{\text{block max},i})_j$ are the $j$-th dimensions of the minimum and maximum vectors of block $i$, respectively.

To select the attended blocks, we adopt a dynamic top-$n$ strategy. First, a fixed number of recent blocks, denoted as $n_{\text{local}}$, are always preserved by setting their scores to $+\infty$, ensuring that the latest context is available for local coherence. Second, we enforce a minimal block number $n_{\text{min}}$ to avoid significant performance degradation on short sequences. Finally, the value of $n$ is dynamically determined based on a active ratio $p$, following:

$$n = \max\left(n_{\text{min}}, \lceil M \times p \rceil\right), \tag{4}$$

where $M$ is the total number of available memory blocks.

## 2.2 DENSE RECTIFICATION

Transformer inference implicitly consists of two distinct phases: **context encoding**, realized through the construction of the KV cache, and **next-token prediction**, realized through the forward pass of the current token. While sparse attention effectively approximates the next-token prediction phase, it inevitably introduces errors. Crucially, these prediction errors accumulate in the KV cache during decoding, leading to compounding inaccuracies over long sequences. To mitigate this issue, we propose **Dense Rectification**, a lightweight mechanism that periodically refreshes the KV cache to maintain its quality. This design constrains error accumulation within a constant window size and enables efficient sparse decoding without compromising generation consistency.

**Rectification Algorithm** Given a rectification frequency $f$, we perform standard sparse decoding for up to $f$ tokens, appending newly generated tokens into the KV cache. After every $f$ token, we batch these recent tokens and re-encode them using dense attention to reconstruct an updated KV cache. This two-phase approach — serial sparse decoding followed by parallel rectification — ensures that errors introduced by approximate attention are corrected at regular intervals, keeping the memory quality close to that of dense decoding. Importantly, the rectification step amortizes efficiently over large batches, maintaining high throughput even when dense recomputation is involved. To maintain consistency, we also refresh the associated block keys during rectification. otherwise, the misalignment between the block keys and the updated KV cache would degrade subsequent sparse retrieval accuracy.

---

**Algorithm 1** Rectified Sparse Decoding

---

**Require:** Initial prompts $\mathcal{P}$, model $\mathcal{M}$, rectification frequency $f$, maximum generation steps $T$
**Ensure:** Generated tokens $\mathcal{G}$
  Initialize KV cache $\mathcal{K}$ by $\mathtt{Prefill}(\mathcal{P}, \mathcal{K})$
  Initialize block key cache $\mathcal{B}$
  Initialize output sequence $\mathcal{G} \leftarrow$ empty
  **for** $i = 1$ **to** $T$ **do**
    $t \leftarrow \mathtt{SparseForward}(\mathcal{G}[i-1], \mathcal{K}, \mathcal{B})$
    Append $t$ to $\mathcal{G}$
    Update KV cache $\mathcal{K}$ with $t$
    Update block key cache $\mathcal{B}$ with $t$
    **if** $i \bmod f = 0$ **then**
      $\mathcal{K}, \mathcal{B} \leftarrow \mathtt{DenseForward}(\mathcal{G}[i-f:i], \mathcal{K}, \mathcal{B})$
      Update block key cache $\mathcal{B}$
    **end if**
  **end for**

---

**Relation to Speculative Decoding** Our rectification strategy is conceptually similar to speculative decoding (Leviathan et al., 2023), as both frameworks balance fast, approximate generation with slower, accurate validation. However, a key difference is that speculative decoding performs accept/reject decisions for each token using a secondary verification model. In contrast, our method directly accepts all tokens generated within a rectification window without any rejection or regeneration. This simplification is effective because sparse attention maintains high predictive quality within reasonably small windows (e.g., $f = 32$ or $f = 64$), thereby avoiding speculative failures and reducing control overhead.

**Compatibility with LLM Serving Systems** Dense Rectification is naturally compatible with modern LLM serving optimizations such as continuous batching (Yu et al., 2022) and chunked prefill (Agrawal et al., 2023; Holmes et al., 2024). Since rectification only requires periodic batched re-encoding, it seamlessly fits into systems that dynamically group decoding and prefill workloads to maximize GPU utilization. By maintaining a fixed rectification frequency per request, our method can operate within the batching and scheduling pipelines without introducing special synchronization barriers or inefficiencies.

### 2.3 DECODING PROCEDURE

Our decoding procedure alternates between sparse decoding and periodic rectification to achieve a balance between efficiency and generation quality. The process begins with a standard dense prefill phase, where the initial prompt is encoded into a complete key-value memory for subsequent decoding. During the decoding phase, tokens are generated sequentially using sparse attention, which restricts memory access to a dynamically selected subset of context blocks. This enables fast autoregressive generation with reduced computational and memory costs.

To correct for approximation errors introduced by sparse attention, we periodically perform rectification. Specifically, after a fixed number of decoding steps, we batch the recently generated tokens and re-encode them using dense attention. This refreshes the key-value memory and ensures that accumulated errors are bounded within a constant window, maintaining memory quality close to dense decoding.

The pipeline continues by alternating between sparse generation and rectification until the generation process completes. The design enables scalable long-context inference while preserving the consistency and reliability of the generated outputs.

**Memory Access Analysis** In each sparse decoding step, the memory access consists of two parts: retrieving block keys for selection, proportional to $\mathrm{mem}(\text{KV cache})/b$, and performing sparse attention, proportional to $\mathrm{mem}(\text{KV cache}) \times p$, where $b$ denotes the block size and $p$ denotes the sparsity ratio. For every $f$ steps, a dense rectification is performed, whose amortized cost per step is

$\mathrm{mem}(\text{KV cache})/f$. Therefore, the average memory access per decoding step is approximated as:

$$\mathrm{Avg}(\mathrm{mem}) = \mathrm{mem}(\text{KV cache}) \times \left( \frac{1}{b} + p + \frac{1}{f} \right).$$

Compared to dense decoding, which requires accessing the entire KV cache at every step, our design achieves a theoretical memory access reduction factor of $\frac{1}{b} + p + \frac{1}{f}$. By adjusting $b$, $p$, and $f$, the pipeline can flexibly trade-off between memory efficiency and generation fidelity.

## 2.4 KERNEL IMPLEMENTATION

We develop a custom kernel optimized for the decoding phase, following a split-execution strategy similar to Flash Decoding and incorporating shared KV fetching techniques (Yuan et al., 2025). The key design principle is to assign each GQA group to an individual streaming multiprocessor (SM), ensuring efficient resource utilization and minimal inter-SM communication.

The decoding workload is $\mathrm{batch\_size} \times \mathrm{num\_kv\_heads}$. Given the total number of SMs available on the GPU, the workload is split accordingly to balance the computation between SMs. The splitting is performed at the level of block indices. For each decoding step, a batch of queries typically activates $k$ memory blocks. We evenly partition $k$ active blocks among the available SMs, so that each SM is responsible for approximately $k/\mathrm{split}$ blocks. Each SM independently fetches the required KV entries corresponding to its assigned blocks and performs sparse attention locally. The kernel implementation is described in Section A.

The design achieves high decoding throughput by minimizing memory contention, maximizing SM occupancy, and fully exploiting intra-GQA key sharing during sparse decoding.

## 3 EXPERIMENTS

We evaluate ReSA from different perspectives. First, we make test-time scaling inference on math reasoning tasks (Section 3.1). Second, we simulate inference-time attention pattern on language modeling (Section 3.2). Third, we verify the effectiveness on retrieval (Section 3.3) tasks. Fourth, we analyze the inference advantages (Section 3.4, including kernel-level and end-to-end accelerations.

We choose Qwen2.5 (Yang et al., 2024), a widely-used standard Transformer pre-trained model as evalutaion architectures. We apply ReSA on all of the layers, rather than skipping the first two layers in Quest (Tang et al., 2024). The block size is 16 and the minimal selected block number is $n_{\min} = 16, n_{\mathrm{local}} = 1$ to avoid performance degradation in short context. For longer sequences, the default sparsity ratio is $p = 0.9$. The default rectification frequency is $f = 32$.

## 3.1 LONG REASONING

We evaluate test-time scaling performance on math reasoning tasks. The validation datasets include Minerva Math (Lewkowycz et al., 2022), Gaokao 2023 En (Liao et al., 2024), Olympiad-Bench (He et al., 2024), AIME24, and AMC23. We exclude some well-known math datasets such as GSM8K (Cobbe et al., 2021), and MATH (Hendrycks et al., 2021) since these datasets' average inference length is below 512. We choose DeepSeek-R1-Qwen-Distill 7B (Guo et al., 2025) as the evaluation model. The number of attention head is 28 and KV head is 4. The hidden size is 3584 and the number of layers is 28.

Table 1 shows that while ReSA achieves performance comparable to the dense baseline. In contrast, block-sparse decoding without dense rectification ("Block Sparse") consistently underperforms dense attention. Because StreamingLLM (Xiao et al., 2023) and H2O (Zhang et al., 2023) are query-independent sparse patterns, their results are large behind "Block Sparse", showing the importance of being content-aware. ReSA maintains near-lossless performance in long-context reasoning tasks, whereas Sparse Decoding leads to performance degradation as decoding progresses. Following Quest (Tang et al., 2024), "Block Sparse$_{\mathrm{dense2}}$" in Table 1 applies dense attention to the first two Transformer layers. It shows that manually enforcing dense layers for the first two layers does not result in a significant improvement in math-reasoning tasks.

| | Minerva | Gaokao2023En | OlympiadBench | AIME24 | AMC23 | Avg |
|---|---|---|---|---|---|---|
| *R1-Qwen-Distill 1.5B* | | | | | | |
| Dense | 28.7 | 71.6 | 40.8 | 27.4 | 65.6 | 46.82 |
| StreamingLLM | 2.9 | 11.2 | 1.8 | 0.0 | 1.9 | 3.56 |
| H2O | 2.6 | 14.2 | 3.3 | 0.0 | 4.4 | 4.90 |
| Block Sparse | **29.0** | 67.9 | 38.7 | 21.3 | 60.6 | 43.50 |
| ReSA | 28.1 | **71.8** | **39.5** | **23.0** | **65.4** | 45.56 |
| Avg Length | 6390.8 | 4915.8 | 8991.6 | 12126.4 | 7866.4 | 8058.2 |
| *R1-Qwen-Distill 7B* | | | | | | |
| Dense | 40.4 | 73.8 | 52.3 | 48.1 | 89.0 | 60.72 |
| StreamingLLM | 7.2 | 23.1 | 4.1 | 0.2 | 3.8 | 7.68 |
| H2O | 2.9 | 13.9 | 3.1 | 0.0 | 6.3 | 5.24 |
| Block Sparse | 38.1 | 72.9 | 48.4 | 46.1 | 83.1 | 57.72 |
| Block Sparse$_{\text{dense2}}$ | 37.9 | 72.5 | 48.8 | 44.6 | 83.1 | 57.38 |
| ReSA | **39.7** | **73.5** | **52.3** | **51.1** | **86.0** | 60.52 |
| Avg Length | 4018.7 | 2889.9 | 7520.0 | 10474.5 | 5732.2 | 6127.1 |

Table 1: Performance comparison on math reasoning tasks. While simple sparse decoding methods show a gap with dense decoding, ReSA achieves near lossless long-sequence generation. We compare with StreamingLLM (Xiao et al., 2023), H2O (Zhang et al., 2023), and "Block Sparse" (i.e., ablating dense rectification). Similar to Quest (Tang et al., 2024), "Block Sparse$_{\text{dense2}}$" uses dense attention for the first two Transformer layers.

## 3.2 LANGUAGE MODELING

We evaluate language modeling performance under simulated sparse decoding patterns. Specifically, we divide each input sequence into two parts. Given a total sequence length $L$, we split it into a prefix of length $L - x$ and a suffix of length $x$. The prefix is processed using dense attention, while the suffix uses sparse attention. Here, $x$ effectively controls the rectification frequency. When $x = L$, it corresponds to the sparse decoding baseline, where no rectifying is performed and the entire sequence is encoded using sparse attention.

We conduct our experiments using long-sequence book data. These texts are typically full-length books, often exceeding 64k tokens, making them well suited for evaluating models' performance on long-range dependency modeling. For each target sequence length, we use the same data and truncate from the left to ensure that the prediction tokens are perfectly aligned across all settings. We report the perplexity computed over the final 32 tokens of each sequence to focus on the model's performance in the later decoding stages.

Figure 4 compares the impact of different rectification frequencies on model perplexity. The setting labeled *Decode Only* corresponds to the case where all KV cache entries are generated using dense attention, and sparse attention is only used for decoding. This serves as the upper bound for ReSA. We observe that ReSA significantly reduces the performance gap between dense and sparse decoding. Notably, when $x = 32$, the model's performance almost approaches the upper bound, demonstrating the effectiveness of rectification in mitigating the error accumulation issue inherent in sparse decoding.

In Figure 5, we further examine the effect of different sparsity ratios under a fixed rectification frequency of $x = 32$. We find that there is a noticeable performance gap between the $p = 0.98$ and $p = 0.95$. Although $p = 0.8$ sparsity achieves perplexity comparable to the dense setting, we adopt $p = 0.9$ as the default due to its better trade-off between performance and efficiency. Additionally, since effective block selection strategies can lead to higher achievable sparsity, our method can be further combined with advanced attention selection mechanisms such as SeerAttention (Gao et al., 2024) to enhance runtime efficiency.

## 3.3 LONG-SEQUENCE RETRIEVAL

We conduct experiments on the RULER benchmark to further evaluate the impact of different sparsity levels. Unlike the long-sequence generation tasks, where rectification plays a critical role in mitigating

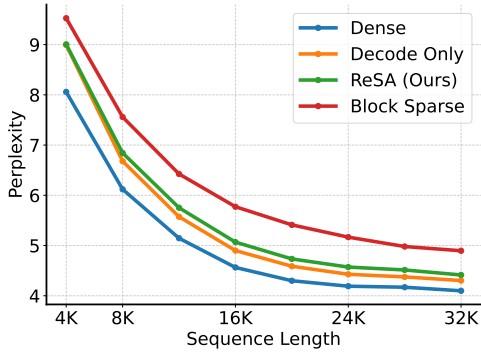 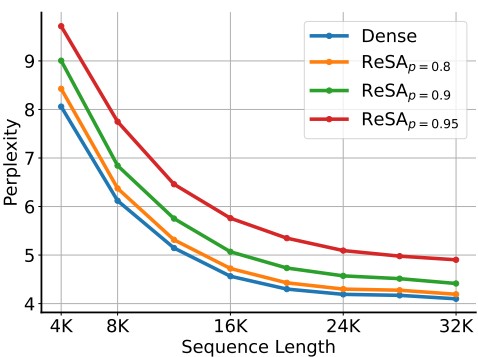

Figure 4: Language modeling perplexity with different rectification frequency.

Figure 5: Language modeling perplexity with different sparsity ratio.

| Setting | QA | MultiQuery | FWE | VT | MultiKey | MultiValue | CWE | Single | Avg |
|---|---|---|---|---|---|---|---|---|---|
| Dense | 0.563 | 0.211 | 0.833 | 0.719 | 0.688 | 0.246 | 0.134 | 1.000 | 0.549 |
| ReSA$_{p=0.95}$ | 0.500 | 0.180 | 0.740 | 0.719 | 0.750 | 0.238 | 0.125 | 1.000 | 0.531 |
| ReSA$_{p=0.9}$ | 0.625 | 0.203 | 0.760 | 0.719 | 0.750 | 0.234 | 0.178 | 1.000 | 0.559 |
| ReSA$_{p=0.8}$ | 0.594 | 0.195 | 0.771 | 0.719 | 0.719 | 0.246 | 0.175 | 1.000 | 0.552 |

Table 2: RULER benchmarks under different sparsity ratios. Dense represents the fully-attended baseline, while ReSA$_{p=x}$ denotes our method with sparsity level $x$.

cumulative error, the RULER benchmark focuses on relatively short output sequences. As a result, the final accuracy is primarily determined by the quality of the sparse attention estimation.

Table 2 shows the long-sequence retrieval results. We observe that as the sparsity ratio increases from $p = 0.95$ to $p = 0.9$, there is a consistent improvement in average accuracy, with ReSA$_{p=0.9}$ achieving comparable performance to the dense baseline (0.559 vs. 0.549). The performance under $p = 0.8$ remains similar to that under $p = 0.9$, indicating that moderate increases in sparsity do not substantially degrade accuracy in short-generation settings. Considering that a lower sparsity ratio generally leads to faster inference, ReSA$_{p=0.9}$ represents a better trade-off between performance and efficiency on the RULER benchmark.

### 3.4 INFERENCE EFFICIENCY

We evaluate the efficiency of ReSA on standard GPU hardware. Specifically, we use Qwen-3 1.7B as the evaluation model and conduct all experiments on NVIDIA H100-80G GPUs. The primary baseline is FlashAttention, a highly optimized dense attention implementation. To ensure a fair comparison and prevent memory overflow issues caused by excessively large KV caches during long-sequence evaluation, we adopt a shared KV cache strategy across all layers during efficiency measurements. The batch size is fixed at 16 by default throughout all experiments.

We integrate ReSA into Nano-vLLM (GeeeekExplorer, 2025), a simplified inference engine which shows similar decoding efficiency as standard vLLM (Kwon et al., 2023). For latency measurement, we report average decoding time. CUDA graph capture is enabled to reduce the CPU overhead.

#### 3.4.1 ATTENTION EFFICIENCY

Figure 6 shows the detailed latency breakdown across different sequence lengths (16k, 64k, and 256k tokens). We compare ReSA, and dense attention under the same settings. The latency is decomposed into three parts: sparse estimation, attention computation, and rectification overhead.

Compared to dense attention, ReSA significantly reduces the total latency, especially at longer sequence lengths. As the sequence grows, dense attention exhibits longer latency with increasing

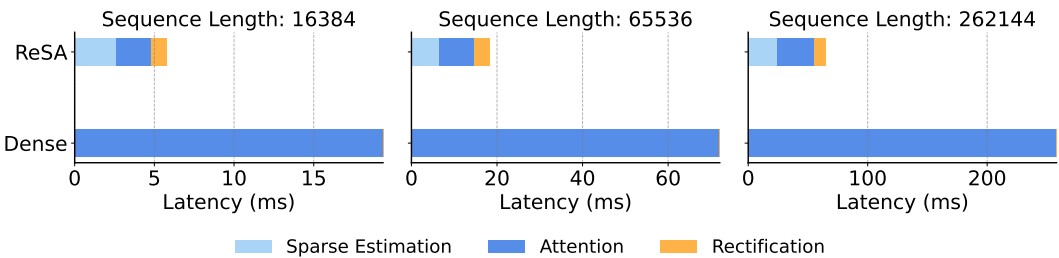

Figure 6: Kernel-level latency breakdown across different sequence lengths. While sparse decoding achieves effective acceleration, rectification only requires a small additional overhead.

context length, leading to substantial latency increase, while ReSA maintains much flatter scaling due to its sparsified attention computation.

Moreover, sparse estimation and attention computation consume comparable amounts of time, because the memory access pattern for sparse estimation scales with $\text{mem}(\text{KV cache})/\text{block}$, while for attention it scales with $\text{mem}(\text{KV cache}) \times p$. Given our experimental settings (block = 16, $p = 0.9$), both operations operate on similar memory volumes. Notably, under fixed block size, further increasing the sparsity ratio can not bring significant speed-up.

The overhead of rectification is relatively small compared with sparse decoding part. Specifically, the rectification module accounts for up to 14.0% of the total attention-related latency at 256k lengths, while at 64k, this proportion drops to 16.6%. When the sequence length is scaling, the latency ratio will converge to the memory access ratio $1/f$. These results indicate that while sparse estimation and attention computation remain efficient, the rectification does not bring big overhead.

### 3.4.2 END-TO-END INFERENCE SPEEDUP WITH NANO-VLLM

Figure 7 reports the end-to-end throughput of LLMs with dense attention and ReSA. The results are estimated under the Nano-vLLM (GeeeekExplorer, 2025) inference framework, so the evaluation setup is similar to real-world deployment. We evaluate the throughput across different context lengths (4K, 16K, 64K, and 256K tokens).

The results are consistent with the kernel-level evaluation as presented in Section 3.4.1. ReSA significantly improves overall throughput as the sequence length grows, achieving up to $3.77\times$ speedup over dense attention. In particular, the benefits of ReSA become more prominent in longer sequences due to the quadratic scaling bottleneck of dense attention, while the overhead of sparse estimation and rectification remains modest even under quantized inference. These results demonstrate that ReSA is highly effective in improving the real-world end-to-end generation speed.

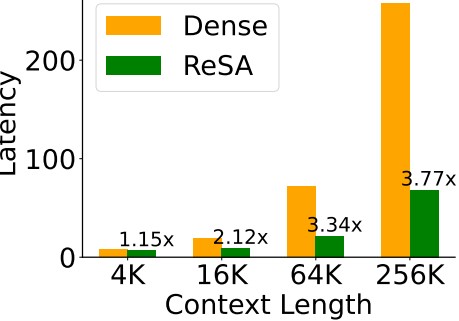

Figure 7: End-to-end latency of LLMs using dense attention and ReSA at various context length. The results are estimated with Nano-vLLM (GeeeekExplorer, 2025), which simulates the product deployment setup.

### 3.5 ABLATION STUDIES

We conduct ablation studies to examine the effect of rectification frequency and sparsity ratio on performance. As shown in Figure 8, we evaluate ReSA across five math reasoning benchmarks under varying sparsity levels ($p \in \{0.9, 0.95, 0.98\}$) and rectification frequencies ($f \in \{16, 32, 64, 128\}$).

Compared to the sparse decoding baseline, ReSA consistently outperforms the baseline across all sparsity levels. Notably, when the attention computation ratio is reduced to 0.1, ReSA achieves accuracy that is remarkably close to the dense decoding upper bound. This demonstrates that ReSA

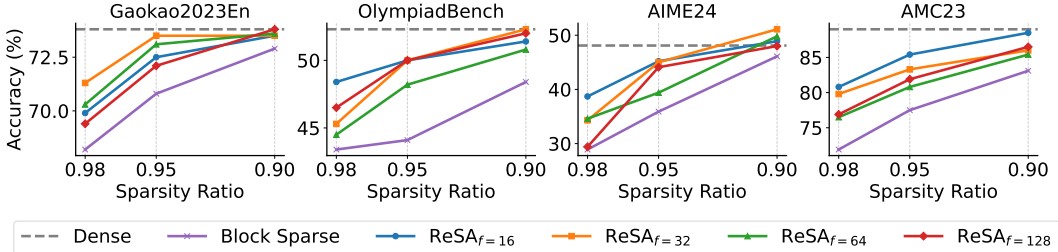

Figure 8: Ablation studies on different rectification frequencies $f$ and sparsity ratios $p$ across five math reasoning benchmarks. ReSA consistently improves over the sparse baseline. Frequencies $f = 32$ or $f = 64$ achieve the best trade-off between performance and overhead.

effectively mitigates the quality drop typically associated with sparse decoding while maintaining high computational efficiency.

Among the frequencies, $f = 32$ achieves accuracy close to the dense baseline on most datasets, striking a favorable balance between quality and efficiency. While $f = 16$ offers marginal gains, it incurs higher rectification overhead and is therefore less practical. Notably, even with $f = 128$, a large portion of the performance gain is retained, highlighting the robustness of the rectification mechanism under infrequent updates.

## 4 RELATED WORK

**Sparse Attention** Recent efforts in sparse decoding for large language models can be broadly categorized into training-free and training-aware approaches. Training-free methods enhance inference efficiency without substantial retraining. Quest (Tang et al., 2024) and InfLLM (Xiao et al., 2024) both adopt query-aware block-sparse attention, selectively retrieving critical memory blocks based on query relevance. MagicPig (Chen et al., 2024) and ClusterKV (Tactic) (Liu et al., 2024) employ similarity-based techniques, using hashing or clustering to approximate attention relevance. In contrast, training-aware architectures such as NSA (Yuan et al., 2025) and MoBA (Lu et al., 2025) integrate sparsity into model design, aligning structures with hardware during pretraining. Our method complements training-free sparse attention by improving memory quality through lightweight rectification, avoiding the high retraining cost required by training-aware approaches.

**Speculative Decoding** Speculative decoding (Leviathan et al., 2023) accelerates generation by drafting multiple tokens and verifying them with the target model. Methods like Medusa (Cai et al., 2024) and EAGLE (Li et al., 2024) reuse the hidden states of the target model for drafting. TriForce (Sun et al., 2024) and MagicDec (Sadhukhan et al., 2024) propose self-speculation, using the model's own sparse KV cache for drafting and a dense cache for verification. Although they share similar compute characteristics with sparse KV-based self-speculation, the methods are orthogonal and complementary. In comparison, self-speculation uses sparse attention for drafting rather than generating final output. Moreover, ReSA does not have per-token accept/reject decisions and resampling overhead. So ReSA is about two times faster than sparse KV-based self-speculation as discussed in Appendix B.

## 5 CONCLUSION

We introduced Rectified Sparse Attention, a simple yet effective method for efficient long-sequence generation. ReSA combines group block sparse attention for decoding latency, and dense rectification to bound error accumulation. Extensive experiments on math reasoning and language modeling tasks show that ReSA achieves near-lossless performance compared to dense decoding. After integrating into Nano-vLLM, ReSA can still deliver up to $3.77\times$ end-to-end inference speedup at 256K context length. These results highlight ReSA's practical effectiveness in long-context language model deployment. By providing a practical, training-free solution that maintains accuracy, ReSA significantly advances the feasibility of deploying large language models for reliable and efficient long-context generation.

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

## A  PSEUDO CODE OF FLASH DECODING KERNEL

The proposed group block sparse attention (Section 2.1) can be easily integrated into the Flash Decoding (Dao et al., 2023) kernel implementation. The modified parts are highlighted as follows.

---

**Algorithm 2** Flash Decoding with Block-Sparse Attention

---

**Require:** Queries $Q$, Keys $K$, Values $V$, block_indices
**Ensure:** Attention outputs $Out_{partial}, logsum_{partial}, Out$
 1: **for** Grid indexed by (num_splits, num_kv_heads, batch_size) **do**
 2:   Load query vectors $q$ in a GQA group
 3:   Compute partial_block_indices with block_indices and num_splits
 4:   Initialize accumulators: $m_i \leftarrow -\infty$, $l_i \leftarrow 1.0$, $acc \leftarrow 0$
 5:   **for** block_id in partial_block_indices **do**
 6:     Load keys $k$ and values $v$ from KV cache in block block_id
 7:     Compute scaled attention scores $qk \leftarrow (qk) \times sm\_scale$
 8:     Apply masking to invalid positions $(qk \leftarrow -1e6)$
 9:     Compute and update $m_i, l_i, acc$
10:   **end for**
11:   Store partial logsum and attention outputs into $logsum_{partial}, Out_{partial}$
12: **end for**
13: Combine different splits Combine($logsum_{partial}, Out_{partial}, Out$)
14: **return** Attention output tensor Out

---

## B  COMPARISON WITH SELF-SPECULATION

As discussed in Section 4, ReSA shares similar computational characteristics with sparse KV cache-based self-speculation. The rectification phase in ReSA resembles the verification phase used in self-speculative methods. However, unlike these methods, ReSA does not rely on output logits to make per-token accept / reject decisions. This design choice is motivated by the observation that, when sparse attention achieves high generation quality, this kind of token-wise strict verification can significantly increase latency without providing proportionate accuracy gains.

To validate this, we compare ReSA and sparse KV-based self-speculation on mathematical reasoning tasks. We set the speculation length to 16, meaning that the model drafts 16 tokens using the sparse KV cache. Similarly, we set ReSA's rectification frequency to 16. Across all tasks, ReSA achieves nearly 2× speedup over self-speculation while maintaining comparable accuracy. This is because, in each verification step of speculative decoding, only about 8 tokens are typically accepted—effectively halving the generation rate compared to ReSA. Although this strict verification ensures that speculative decoding matches the accuracy of dense attention, we have previously shown that ReSA also approaches the accuracy of dense attention. Therefore, we believe that the marginal accuracy gains of speculative decoding do not justify its substantial latency overhead.

| Task | Sparse KV Self-Spec. | Rectified Sparse Attention |
|------|----------------------|----------------------------|
| Minerva | 1× | 1.93× |
| Gaokao2023En | 1× | 1.87× |
| OlympiadBench | 1× | 1.98× |
| AIME24 | 1× | 1.96× |
| AMC23 | 1× | 1.86× |
| **Average** | 1× | 1.92× |

Table 3: Decoding speedup comparison. We set the throughput of self-speculation as baseline. ReSA achieves larger speedup compared with sparse self-speculative decoding.

