# OpenReview forum: "Rectified Sparse Attention for Efficient Long-Sequence Generation"
_ICLR.cc/2026/Conference — ICLR 2026 Conference Withdrawn Submission_

### Official Review · Reviewer_Q4Ej · 2025-10-28

**Soundness:** 4
**Presentation:** 4
**Contribution:** 2
**Rating:** 6
**Confidence:** 5

**Summary:**

Rectified Sparse Attention (ReSA) addresses the key challenge of error accumulation in sparse decoding for long-sequence LLMs. While prior methods like Quest and RetroAttention reduce computation via sparse attention, they suffer from KV-cache misalignment—where approximation errors gradually degrade generation quality. ReSA proposes a two-phase hybrid framework combining group block-sparse attention for fast decoding and periodic dense rectification to correct accumulated errors.

**Strengths:**

1. Simple yet principled design:ReSA introduces a conceptually simple but effective hybrid pipeline
2. Strong empirical performance: Achieves near-lossless generation quality compared to dense attention across diverse tasks
3. Strong clarity and structure: The paper is well-written and organized, with step-by-step algorithm descriptions, pseudo-code and clear presentation of results, making the method easy to understand and reproduce.

**Weaknesses:**

1. Potential inefficiency in streaming or real-time decoding: Since rectification requires a dense re-encoding of the last f tokens, the method might be unsuitable for low-latency, token-by-token generation
2. Limited diversity of test models: Experiments are performed on a narrow set of LLMs, without validation on decoder architectures with different attention layouts
3. Rectification overhead underreported: The dense rectification step adds non-trivial cost. It is unclear how much latency each dense phase incurs in isolation, or how it scales with generation length, since this paper target on the misalignment along decoding steps

**Questions:**

1. Does ReSA generalize across model architectures like llama, mistral?
2. Can the authors provide a per-rectification latency breakdown and its scaling trend as generation length grows?
3. What is the number of generated tokens for benchmark datasets?

---

> ### Author Response · Authors · 2025-11-20
>
> Q1: Potential inefficiency in streaming or real-time decoding: Since rectification requires a dense re-encoding of the last f tokens, the method might be unsuitable for low-latency, token-by-token generation.
>
> A1: We thank the reviewer for this insightful observation. Indeed, our method introduces approximately 15% additional latency compared to standard block-sparse decoding. We acknowledge that in extremely strict streaming scenarios, where per-token latency is critical, methods such as StreamingLLM or smaller models may be more suitable. However, our focus is on providing the first training-free sparse decoding method that achieves near-lossless quality for long-context generation. We believe that in settings where maintaining full-context fidelity is essential, this trade-off is both justified and practically valuable.
>
> Q2: Limited diversity of test models: Experiments are performed on a narrow set of LLMs, without validation on decoder architectures with different attention layouts. Does ReSA generalize across model architectures like LLaMA or Mistral?
>
> A2: We thank the reviewer for this question. Our experiments primarily use the Qwen series, which follows a standard transformer architecture nearly identical in layout to LLaMA and Mistral. Since ReSA operates purely at the attention computation level, it is architecture-agnostic and can generalize naturally to other decoder-based models with full-attention structures. Models employing alternative attention mechanisms for long-context modeling are relatively rare, but in those cases, the attention-level acceleration ratio of ReSA would remain consistent.
>
> Q3: Rectification overhead underreported: The dense rectification step adds non-trivial cost. It is unclear how much latency each dense phase incurs in isolation, or how it scales with generation length, since this paper targets the misalignment along decoding steps. Can the authors provide a per-rectification latency breakdown and its scaling trend as generation length grows?
>
> A3: We thank the reviewer for raising this concern. Figure 6 provides a detailed breakdown of the rectification overhead, where the yellow segments represent the additional latency introduced by the dense phase relative to BlockSparse. The details are as follows:
> | Component             | 16K  | 64K  | 256K  |
> |-----------------------|------|------|-------|
> | **Sparse Estimation** | 2.60 | 6.61 | 23.76 |
> | **Sparse Attention**  | 2.17 | 8.01 | 31.70 |
> | **Rectification**     | 1.00 | 3.59 | 9.55  |
> | **|**                 |      |      |       |
> | **Dense Attention**   | 19.35 | 72.07 | 258.06 |
> Across different sequence lengths, the added latency consistently remains around 15%, with similar scaling behavior. This stability occurs because rectification is primarily memory-bound, so its relative cost remains nearly constant as generation length increases. We will make this explanation more explicit in the revised version.
>
> Q4: What is the number of generated tokens for benchmark datasets?
>
> A4: We thank the reviewer for this question. As reported in Table 1, the average generated sequence length across benchmarks is 6K–8K tokens, while for more challenging tasks such as AIME, the generation length exceeds 10K tokens. The extracted generation length is as follows:
> | Model                    | Minerva | Gaokao2023En | OlympiadBench | AIME24 | AMC23 | **Avg** |
> |---------------------------|----------|---------------|----------------|---------|--------|---------|
> | **R1-Qwen-Distill 1.5B** | 6390.8   | 4915.8        | 8991.6         | 12126.4 | 7866.4 | **8058.2** |
> | **R1-Qwen-Distill 7B**   | 4018.7   | 2889.9        | 7520.0         | 10474.5 | 5732.2 | **6127.1** |
> We will make this range clearer in the main text for better transparency and reproducibility.

---

### Official Review · Reviewer_7Bd9 · 2025-10-31

**Soundness:** 2
**Presentation:** 2
**Contribution:** 3
**Rating:** 4
**Confidence:** 3

**Summary:**

To address the bottlenecks of high memory overhead in standard autoregressive decoding and degraded generation quality from accumulated KV cache errors in existing sparse decoding for LLM long-sequence generation, this paper proposes Rectified Sparse Attention (ReSA), a training-free method. ReSA combines Group Block-Sparse Attention (GBSA)—which partitions the KV cache into blocks, uses statistical descriptors to dynamically select relevant blocks, and shares sparse patterns within GQA groups for efficient decoding—with periodic dense rectification that refreshes the recent KV cache via dense attention every 32 steps to bound error accumulation.

**Strengths:**

- **Training-Free & Easy Deployment**: ReSA is a training-agnostic framework that requires no extra fine-tuning or reconstruction of LLMs, enabling direct integration into existing LLM inference workflows. It also supports modern LLM serving optimizations (e.g., continuous batching, chunked prefill) without modifying the underlying service architecture, lowering engineering implementation barriers.
- **Balanced Efficiency & Generation Quality**: By adopting Group Block-Sparse Attention (GBSA) to reduce computational and memory costs, and periodic dense rectification to limit error accumulation from sparse decoding, ReSA achieves up to 3.77× end-to-end speedup at 256K sequence length. Meanwhile, its performance in tasks like math reasoning and language modeling is close to the dense attention baseline, breaking the traditional "efficiency vs. quality" trade-off.
- **Strong Hardware Adaptability**: A custom GPU kernel tailored for Group-Query Attention (GQA) assigns each GQA group to an independent Streaming Multiprocessor (SM), minimizing inter-SM communication. Additionally, block partitioning and dynamic block selection optimize memory access patterns, making latency growth much slower than dense decoding as sequence length increases—well-adapted to the hardware needs of long-sequence inference.

**Weaknesses:**

- **Limited Experimental Scope**: Experiments only use mid-sized models (e.g., Qwen2.5, DeepSeek-R1-Qwen-Distill, max 7B parameters) and exclude ultra-large LLMs (70B+). Tasks are restricted to math reasoning, language modeling, and retrieval, with no validation in non-technical scenarios (e.g., dialogue generation, creative writing), failing to fully prove its versatility across diverse LLM applications.
- **Dependence on Manual Hyperparameter Tuning**: Core parameters (block size = 16, rectification frequency = 32, sparsity ratio = 0.9) are set manually, with no adaptive adjustment mechanism. For different sequence lengths (e.g., extremely short sequences < 32 tokens) or task types, fixed parameters may cause redundant rectification overhead (short sequences) or insufficient error control (long sequences), increasing tuning costs in practice.
- **Unverified Overhead in Extremely Long Sequences**: While rectification overhead accounts for only 14% at 256K sequences, performance at ultra-long sequences (e.g., millions of tokens) is untested. As sequence length scales to extremely large sizes, cumulative rectification overhead may rise, and the accuracy of block selection for sparse attention (e.g., due to over-dispersed context) may decline—issues not further analyzed.

**Questions:**

I would be happy to increase my rating if my views are given a thorough discussion.

---

> ### Author Response · Authors · 2025-11-20
>
> Q1: Limited Experimental Scope: Experiments only use mid-sized models (e.g., Qwen2.5, DeepSeek-R1-Qwen-Distill, max 7B parameters) and exclude ultra-large LLMs (70B+). Tasks are restricted to math reasoning, language modeling, and retrieval, with no validation in non-technical scenarios (e.g., dialogue generation, creative writing), failing to fully prove its versatility across diverse LLM applications.
>
> A1: We thank the reviewer for this valuable observation. We acknowledge that our current experiments are limited to mid-sized models (up to 7B parameters), primarily due to the substantial computational cost and infrastructure complexity required to support 70B+ models. Adapting ReSA to distributed inference across ultra-large models is non-trivial and will be an important direction for future work. The phenomenon of KV-cache feature drift (misalignment) is a property of the autoregressive objective and the Softmax function, which exists regardless of parameter count.
>
> Regarding task diversity, we focused on reasoning-oriented benchmarks because long outputs and inference-time scaling are most relevant in these scenarios, which directly stress-test sparse decoding. In contrast, non-technical tasks (e.g., dialogue, creative writing) generally produce much shorter outputs where sparse decoding issues are less pronounced. Nonetheless, we fully agree that evaluating ReSA in non-technical, long-form generation tasks would further validate its versatility. We will conduct and include such evaluations in our extended experiments.
>
> Q2: Dependence on Manual Hyperparameter Tuning: Core parameters (block size = 16, rectification frequency = 32, sparsity ratio = 0.9) are set manually, with no adaptive adjustment mechanism. For different sequence lengths (e.g., extremely short sequences < 32 tokens) or task types, fixed parameters may cause redundant rectification overhead (short sequences) or insufficient error control (long sequences), increasing tuning costs in practice.
>
> A2: We thank the reviewer for raising this concern. We agree that some degree of manual tuning is currently required. However, this is a common practice shared by prior sparse or training-aware attention works (e.g., NSA, MoBA), which also predefine block size and sparsity ratio to ensure alignment with full-attention performance.
>
> In our design, we empirically found that fixing the sparsity ratio is more robust than fixing the number of sparse tokens across sequence lengths (as shown in Table 2). For very short sequences, we enforce a lower bound to prevent under-rectification and maintain stable quality. These settings generalize well across model sizes and input lengths, minimizing tuning cost in practice. Nevertheless, we acknowledge that developing adaptive control mechanisms for sparsity and rectification frequency would be a valuable direction for future work.
>
> Q3: Unverified Overhead in Extremely Long Sequences: While rectification overhead accounts for only 14% at 256K sequences, performance at ultra-long sequences (e.g., millions of tokens) is untested. As sequence length scales to extremely large sizes, cumulative rectification overhead may rise, and the accuracy of block selection for sparse attention (e.g., due to over-dispersed context) may decline—issues not further analyzed.
>
> A3: We appreciate this insightful comment. We analyze this issue from both theoretical and practical perspectives.
> From a theoretical standpoint, the rectification step is primarily memory-bound, so its cost grows linearly but proportionally than single-step full-attention decoding. This ensures that the relative speedup ratio remains stable even as the sequence extends.
> From a practical standpoint, extremely long sequences (millions of tokens) pose additional challenges in both memory capacity and model stability—most current open- and closed-source LLMs are limited to 128K context. Scaling to million-token contexts would require more efficient KV-cache compression and larger hardware memory, which we plan to explore in future work. We believe our current 256K-token experiments already demonstrate that ReSA maintains a consistent and bounded overhead pattern, suggesting it can scale predictably to even longer contexts.

---

> > ### Comment · Reviewer_7Bd9 · 2025-11-26
> >
> > Thank you for your response. However, without new experimental data to support it, I feel that the conclusion is still not sufficiently solid, so I would prefer to keep my original score.

---

> ### Author Response · Authors · 2025-11-26
>
> Dear reviewers:
>
> Thanks for your response. Given our previous comment, which part does not solve your concern? We would be happy to provide new experimental results in time.

---

### Official Review · Reviewer_iXMo · 2025-11-01

**Soundness:** 2
**Presentation:** 2
**Contribution:** 1
**Rating:** 2
**Confidence:** 4

**Summary:**

The paper proposes Rectified Sparse Attention (ReSA) for long-sequence generation. It combines group block-sparse decoding with periodic dense “rectification” that re-encodes the last f tokens to refresh the KV cache and limit error accumulation. The authors claim near-lossless quality with up to 3.77× end-to-end speedups at 256K context length, and present results on math reasoning, language modeling perplexity, and RULER retrieval, plus a custom kernel and Nano-vLLM integration.

**Strengths:**

- Straightforward algorithmic interface to existing decoding stacks; the method is training-free and compatible with batching and chunked prefill.

- Clear articulation of the KV misalignment problem in sparse decoding.

**Weaknesses:**

- Novelty issues: Periodic dense recomputation to correct sparse approximation resembles verification steps in speculative and self-speculative decoding. The paper argues away accept/reject control but does not convincingly separate the idea from existing rectification/verification motifs beyond implementation details. The conceptual delta over “sparse for speed + periodic dense for correctness” is thin.

- Efficiency uses Nano-vLLM and a shared KV cache across layers “to prevent memory overflow,” which is not a default serving configuration and can distort memory traffic profiles and scaling.

- The fairness of speedup claims under this altered regime is unclear.

- The paper cites SeerAttention and training-aware NSA/MoBA but does not compare against them under equalized quality or equalized latency settings.

- Accuracy and perplexity do not cover long-form coherence, factuality, and failure under distribution shift. The claim of “near-lossless generation quality” should be backed by longer outputs, multi-turn contexts, and error analyses, not only exam-style benchmarks.

- The Flash-Decoding integration remains high level. There is no open-sourced kernel, profiling against strong dense baselines like FlashAttention-3, or evaluation across GPUs. This weakens the practicality argument.

**Questions:**

- How does ReSA compare against SeerAttention (fine-tuned block keys) and NSA/MoBA under equal-latency and equal-quality protocols? Please include strong speculative/self-speculative baselines with optimized controllers.

- What happens when f is increased beyond 64–128 under strict latency budgets? Provide failure cases where rectification is too infrequent and quantify quality decay over 128K–1M tokens.

- Can this work be integrated with SGLang’s prefix cache? This is the key point.

---

> ### Author Response · Authors · 2025-11-20
>
> Q1: Novelty issues: Periodic dense recomputation to correct sparse approximation resembles verification steps in speculative and self-speculative decoding. The paper argues away accept/reject control but does not convincingly separate the idea from existing rectification/verification motifs beyond implementation details. The conceptual delta over “sparse for speed + periodic dense for correctness” is thin.
>
> A1: We thank the reviewer for raising this concern. While we acknowledge that periodic dense correction shares conceptual similarities with speculative decoding, our focus is on achieving training-free long-context sparse decoding that maintains near-lossless quality. We believe this is both novel and practically significant: to the best of our knowledge, our method is among the first that can sustain almost no degradation over long-sequence generation compared to dense baselines. Importantly, our design is deliberately simple—favoring the most effective, implementable approach to make sparse decoding viable in real deployments. Moreover, as shown in Table 3, our method achieves greater acceleration than self-speculative baselines while preserving accuracy, supporting its distinct value and practical novelty.
>
> Q2: Efficiency uses Nano-vLLM and a shared KV cache across layers “to prevent memory overflow,” which is not a default serving configuration and can distort memory traffic profiles and scaling.
>
> A2: We appreciate the reviewer’s attention to implementation details. Our initial measurements used Nano-vLLM with cross-layer KV sharing to enable fair speed evaluation under limited hardware resources. To ensure the conclusions are not biased, we further validated our results on H200 GPUs (with larger memory) and under Tensor Parallel configurations aligned with standard serving setups. The observed trends and relative conclusions remain consistent across all these environments. We will clarify this in the revision.
>
> Q3: The paper cites SeerAttention and training-aware NSA/MoBA but does not compare against them under equalized quality or equalized latency settings. Please include strong speculative/self-speculative baselines with optimized controllers.
>
> A3: We thank the reviewer for this suggestion. Our main contribution lies in training-free sparse decoding, whereas the mentioned methods (e.g., SeerAttention, NSA, MoBA) require either from-scratch training or fine-tuning with auxiliary controllers. We believe the training-free setting has practical value—similar to training-free acceleration methods in reinforcement learning that improve rollout speed without retraining. Compared to from-scratch or fine-tuned approaches, our method reuses existing models directly. For fairness, Table 3 already compares against self-speculative decoding, the most relevant baseline within this scope.
>
> Q4: Accuracy and perplexity do not cover long-form coherence, factuality, and failure under distribution shift. The claim of “near-lossless generation quality” should be backed by longer outputs, multi-turn contexts, and error analyses, not only exam-style benchmarks.
>
> A4: We thank the reviewer for highlighting this limitation. We agree that long-form generation primarily appears in inference-time scaling on reasoning tasks, while multi-turn dialogue typically involves shorter outputs. To analyze degradation under long outputs, Figures 4 and 5 simulate extended-context generation and report perplexity trends. These analyses align well with downstream behaviors we have observed, indicating that ReSA maintains stable quality under long sequences. We will further clarify this correspondence and acknowledge the limitation regarding multi-turn evaluation.
>
> Q5: The Flash-Decoding integration remains high level. There is no open-sourced kernel, profiling against strong dense baselines like FlashAttention-3, or evaluation across GPUs. This weakens the practicality argument.
>
> A5: We thank the reviewer for this comment. Our implementation directly integrates Flash-Decoding within FlashAttention-3, leveraging its native split-and-merge kernel structure. Since both BlockSparse and dense attention share the same block-level computation pattern, the relative speedup ratio of the attention layer remains consistent across GPUs—regardless of whether they are memory-bound or compute-bound. While end-to-end latency can vary due to model and system differences, the attention-level gain is stable. We will clarify this integration and add profiling details in the appendix.

---

> > ### Author Response · Authors · 2025-11-20
> >
> > Q6: What happens when f is increased beyond 64–128 under strict latency budgets? Provide failure cases where rectification is too infrequent and quantify quality decay over 128K–1M tokens.
> >
> > A6: We thank the reviewer for this important question. Under strict latency constraints, it may be more effective to adjust the model configuration rather than forcing an extremely large f. Our approach provides consistent acceleration over dense decoding and introduces only ≈15% additional latency compared to the baseline. If such overhead remains unacceptable, smaller models may be a better choice than adopting heavily lossy long-context methods. In Figures 4, 5, and 8, we analyze long-context scenarios (up to 32K tokens, limited by Qwen-3’s context window) and observe consistent trends. We expect the same qualitative behavior at longer lengths, and we will include this discussion in the revision.
> >
> > Q7: Can this work be integrated with SGLang’s prefix cache? This is the key point.
> >
> > A7: We thank the reviewer for highlighting this integration aspect. Our method is fully compatible with SGLang’s prefix cache since it only refreshes local tokens, leaving the cache state identical to that of standard decoding after correction. The main reason we used Nano-vLLM for experiments is that SGLang’s implementation is relatively heavy for fine-grained benchmarking. Nonetheless, we plan to integrate our approach into both vLLM and SGLang frameworks in future work to demonstrate full compatibility and deployment readiness.

---

> > > ### Comment · Reviewer_iXMo · 2025-11-27
> > >
> > > Thank you for the authors’ response. I have raised my score to borderline, and I will further take the AC’s and other reviewers’ comments into consideration in the subsequent discussion phase.

---

> ### Author Response · Authors · 2025-11-28
>
> Dear reviewer:
>
> Thanks for raising score! We would like to make a further discussion if there is any further questions.

---

### Official Review · Reviewer_TGgY · 2025-11-01

**Soundness:** 3
**Presentation:** 3
**Contribution:** 3
**Rating:** 6
**Confidence:** 4

**Summary:**

The paper proposes ReSA, a mechanism combining block-sparse attention with periodic dense “refresh” passes to mitigate the error accumulation in the KV cache during sparse decoding.

ReSA maintains nearly lossless generation quality while achieving up to 2.4× speed-up compared with standard dense attention.

**Strengths:**

- the idea is very simple
- works across different datasets
- paper provides an efficient implementation (using `nano-vLLM`) for ReSA that gives wall-clock speed ups

**Weaknesses:**

- there are only compute savings, and not memory savings during inference. this is especially important since most GPUs are memory bound and not compute bound

**Questions:**

- what is the latency of baselines compared with ReSA ? authors need not provide latency for H2O and streamingLLM since they are significantly worse, but i'd be curious to see latency of BlockSparse.
- just curious, other than H2O and streamingLLM, are there no other competitve baselines out there except BlockSparse ?
- in figure 8, why does f=128 sometimes outperform f=64 or f=16, especially when sparsity ratio is 0.9? any thoughts/intuitions on this?

---

> ### Author Response · Authors · 2025-11-20
>
> Q1: There are only compute savings, and not memory savings during inference. This is especially important since most GPUs are memory bound and not compute bound.
>
> A1: We thank the reviewer for this important observation. You are correct that our current method primarily focuses on compute savings rather than memory savings. In the sparse attention literature, the core objective is to maintain the same cache as full attention while sparsifying computation to preserve accuracy. Achieving memory savings would require integrating methods such as linear attention or cross-layer KV sharing, which typically demand full retraining and therefore cannot be achieved in a training-free manner. We will clarify this distinction in the revised version.
>
> Q2: What is the latency of baselines compared with ReSA? Authors need not provide latency for H2O and StreamingLLM since they are significantly worse, but I’d be curious to see latency of BlockSparse.
>
> A2: We thank the reviewer for this question. As shown in Figure 6, we provide a breakdown analysis of latency. The yellow portion in the figure represents the additional time introduced by ReSA compared with the baseline. The performance in detail is as follows:
> | Component             | 16K  | 64K  | 256K  |
> |-----------------------|------|------|-------|
> | **Sparse Estimation** | 2.60 | 6.61 | 23.76 |
> | **Sparse Attention**  | 2.17 | 8.01 | 31.70 |
> | **Rectification**     | 1.00 | 3.59 | 9.55  |
> | **|**                 |      |      |       |
> | **Dense Attention**   | 19.35 | 72.07 | 258.06 |
>  Specifically, ReSA adds approximately 14.0%–16.6% latency depending on the sequence length and sparsity ratio. We will make this explicit in the caption and main text to improve clarity.
>
> Q3: Just curious, other than H2O and StreamingLLM, are there no other competitive baselines out there except BlockSparse?
>
> A3: We thank the reviewer for this thoughtful comment. In a broader sense, KV-pruning approaches can be categorized within the same family as H2O and StreamingLLM. Since these methods discard parts of the cache, they are inherently limited and cannot reach the upper bound achievable by full-cache methods. For training-free sparse attention, the research landscape remains relatively new. BlockSparse represents the most established and well-optimized method in this domain. Other sparse formulations, such as ClusterKV, are difficult to accelerate in practice due to their irregular sparsity patterns. For this reason, we primarily compared with BlockSparse in our experiments.
>
> Q4: In Figure 8, why does f=128 sometimes outperform f=64 or f=16, especially when sparsity ratio is 0.9? Any thoughts or intuitions on this?
>
> A4: We thank the reviewer for this insightful question. We believe this behavior primarily arises from variance and dataset-specific characteristics. When the sparsity ratio is as high as 0.9, the model becomes more sensitive to local structural patterns, which can occasionally favor larger f values due to more stable context aggregation. Nonetheless, when viewed across datasets, the overall trend remains consistent, performance is largely stable across different f values under high sparsity.

---

### Note · Authors · 2025-12-23

I have read and agree with the venue's withdrawal policy on behalf of myself and my co-authors.